# Empowerment and utilization of HIV testing among partnered women in Zambia: Evidence from the Zambia demographic and health survey 2018

**Whiteson Mbele**[1,2]*, **Phyllis Dako-Gyeke**[1], **Andreas Ndapewa Frans**[1], **Jean Claude Ndayishimiye**[1], **Jordanne Ching**[1]

**1** Department of Social and Behavioral Sciences, School of Public Health, College of Health Sciences, University of Ghana, Accra, Ghana, **2** Kasiya Mission Hospital, Pemba District Health Office, Pemba, Southern Province, Zambia

* whitesonmbele@gmail.com

## Abstract

In Zambia, women are disproportionally more affected by HIV compared to men. This has mainly been attributed to harmful gender norms that enhance male dominance and disempower women, preventing them from exercising their right to negotiate for safe sex and utilizing HIV prevention services such as HIV testing. This study examined associations between empowerment and HIV testing among married and partnered women. We analyzed secondary data from the couple's recode of the 2018 Zambia demographic and health survey. Univariable and multivariable logistic regression analysis was conducted, and p<0.05 was considered statistically significant. We included a total of 5,328 married and partnered women in the analysis, of which 5057 (94.9%) had undergone an HIV test before. After adjusting for confounders, decision-making was the only independent predictor of HIV testing among measures of empowerment. Women who were highly empowered in decision-making were more likely to have undergone an HIV test compared to those who were less empowered (AOR = 2.1; 95% CI: 1.5, 2.9). Women aged 20–29 years (AOR = 2.4; 95% CI: 1.6, 3.6), 30–39 years (AOR = 5.3; 95% CI: 3.4, 8.2), or 40–49 years (AOR = 2.9; 95% CI: 1.9, 4.7), those with primary education (AOR = 2.4; 95% CI: 1.7, 3.4) or secondary and higher (AOR = 4.1; 95% CI: 2.3, 7.2), rich women (AOR = 2.4; 95% CI: 1.5, 3.7) or women with middle wealth (AOR = 1.5; 95% CI: 1.1, 2.2) and those who gave birth in the last 5 years (AOR = 3.3; 95% CI: 2.5, 4.5) were more likely to have been tested for HIV. This study highlights the critical influence of women's empowerment in decision-making on HIV testing. Additionally, level of education, wealth, age, and having given birth before are essential factors to consider in promoting HIV testing among women in Zambia.

**Data Availability Statement:** For this analysis, we used the 2018 Zambia demographic and health

survey data set. The data is easily accessed for free from The DHS Program website at (https://dhsprogram.com/data/available-datasets.cfm) upon request. We do not hold any special access privileges to this data.

**Funding:** The authors received no specific funding for this work.

**Competing interests:** The authors have declared that no competing interests exist.

## Introduction

Of the estimated 39 million people living with HIV and AIDS globally in 2022, more than half (53%) were Women and girls [1]. In Sub-Saharan Africa, over 63% of all new HIV infections occur among Women and girls [1]. Women's access to HIV prevention services, such as HIV testing is influenced by the level of empowerment [2, 3]. Women who are empowered can make independent decisions regarding their health, including HIV testing without depending on their husbands or partners [4]. Empowered women can negotiate for safer sex with their partners, reducing the chances of risky sexual behaviors [5]. However, harmful gender norms that enhance male dominance prevent women from exercising their right to negotiate for safe sex and utilize HIV prevention services such as HIV testing, putting them at a disproportionally higher risk of HIV transmission [6]. To address gendered risks of HIV transmission, a call for strategies that address female disempowerment has been made [7].

The ability of Women to exercise their sexual and reproductive rights is crucial in achieving gender equity and equality in health. Among married or cohabiting couples, gender norms that limit Women's decision-making power about their sexual and reproductive rights are key drivers of HIV transmission [8, 9]. It is evident from previous studies that a large proportion of HIV infections occur within marital or cohabiting relationships [8–12]. Women are only empowered if they consider themselves to have the right and be able to make independent decisions about their health [13]. However, gender norms that promote Women's subordination and justify men's dominance are key obstacles to Women's empowerment and affect the ability of Women to protect themselves from sexually transmitted infections (STIs), including HIV [14, 15]. In Zambia for example, the cultural expectation of submission and obedience of Women and girls to their partners has promoted male dominance, leaving women with little voice [16]. This has contributed to a disproportion in HIV transmission between boys and girls, with girls and young women carrying the heaviest burden of HIV prevalence (13%) compared to boys and men (7%) [16, 17].

The significance of women's empowerment in addressing the HIV epidemic cannot be overstated. Empowerment plays a pivotal role in instilling confidence among women to undergo HIV testing. This newfound confidence not only allows women to ascertain their HIV status but also empowers them to take preventive measures, such as averting mother-to-child transmission of HIV (MTCT) and minimizing the risk of transmitting the virus to their partners [3]. The extent to which a woman has been empowered significantly influences her accessibility to HIV testing [18, 19]. An empowered woman, whether through cultural, political, or professional means, is more likely to autonomously decide to undergo HIV testing [3]. Such empowerment liberates her from dependency on her husband or partner for decisions related to HIV testing, fostering a proactive approach to her healthcare. In Zambia, HIV testing among pregnant women declined from 85% in 2013 to 82% in 2018 [20]. The underutilization of HIV testing services in Zambia has been attributed to factors such as social relations [21], gender inequalities [22–24] and individual beliefs and perceptions [25]. Promoting joint testing and decision-making among couples enhances communication and mutual support in tackling HIV-related health issues [26]. Couples-based HIV testing is particularly effective for preventing mother-to-child transmission of HIV [26–29]. It encourages women to adhere to antiretroviral therapy (ART) by reducing the need to conceal their HIV status and medication due to fears of intimate partner violence [30–32]. In Zambia, only 10% of couples participate in joint testing, highlighting a critical area for improvement [33].

To increase the ability of women to make independent and joint decisions regarding their health, including HIV testing, Zambia strengthened its policy and legal frameworks for promoting gender equality and women empowerment [34]. The country has observed a positive

drop in the gender inequality index from 0.627 in 2011 to 0.540 in 2021 [34, 35]. Despite this step in the right direction, the country is still ranked low on gender gap reduction and inequalities, sitting at 138 out of 195 countries in the gender inequality index [35]. The slow pace of the attainment of women's empowerment and gender equality in Zambia has been attributed to factors such as the conservation of cultural values that disempower women in decision-making [34]. Empowerment has been variably defined, with limited data for comparisons across countries, especially in Sub-Saharan Africa [36–38]. According to the World Bank definition, empowerment is a process of change by which those who have been denied the ability to make strategic life choices acquire this ability [39]. In the context of the current study, we define women's empowerment as the process of increasing women's ability to make autonomous decisions, exercise control over their lives, and access resources and opportunities to enhance their well-being [40, 41]. Women are considered empowered when they can envision a different life for themselves and feel capable and entitled to make decisions [13]. With the growing need for a reliable indicator to measure and track progress on women and girls empowerment, several empowerment indicators have been proposed and contextualized [36–38, 42]. The Survey-based Women's empowerment Index (SWPER) is one such indicator developed in 2017 for use in Low- and Middle-Income countries in the African context to address the need for a single consistent measure of Women empowerment [42]. The SWPER was validated using Demographic and health survey (DHS) data from 34 African countries and attracted interest from international agencies and the academic community [42]. In 2020, the SWPER was expanded beyond Africa to all low and middle-income countries (LMICs) to make it a global monitoring tool [43]. The indicator measures three domains of empowerment namely, attitude to violence, social independence, and decision making using fourteen questions [43]. The indicator uses individual-level data which allows for the assessment of associations between empowerment and several health interventions or outcomes [42]. Because of its global validation, the SWPER allows for not only within-country comparisons but also between-country comparisons as well as time and trend analysis as new data emerge [43]. Despite the robustness and novelty of the SWPER index in assessing associations between measures of women empowerment and health-related outcomes, the tool has not been utilized in Zambia to unravel the challenge of underutilization of HIV testing services among women in the country.

This study therefore assessed the effects of empowerment on HIV testing among coupled women in Zambia, employing the SWPER index as a measure of empowerment. To the best of our knowledge, this was the first study to utilize the novel, globally standardized SWPER indicator to determine the influence of empowerment on HIV testing among Zambian women. Results from this study will set the baseline for monitoring of progress made on Women empowerment in Zambia and identify areas for priority in HIV testing programs.

## Materials and methods

### Study area

The area for this study was Zambia. The Republic of Zambia is in Southern Africa and is a low-income country. The country has a population of 19,610,769 with a male population of 9,603,056 and a female population of 10, 007,713 according to the 2022 national census [44]. Zambia is one of the countries possessing unequal gender norms, with a Gender Inequality Index (GII) of 0.540 in 2021, ranking 138 out of 195 countries globally [35]. This gender inequity in health has contributed to the observed higher HIV infection among females compared to males in the country [45–49], impeding the achievement of sustainable development goal 5 (SDG 5). Moreover, the country recorded a drop of 43.7% in HIV testing in 2020 compared to

2019, with similar decreases observed across both sexes [50]. It is essential to recognize that multiple factors, beyond gender norms, may have contributed to the observed decline in the HIV testing rate. The country has an ambitious goal of achieving the joint United Nations Program on HIV/AIDS (UNAIDS) "95-95-95" targets which state that by 2030, 95% of people living with HIV should be aware of their status, 95% of those aware of their status should be on antiretroviral therapy (ART), and 95% of those on ART should be virally suppressed [50]. The country made commendable progress in achieving the targets in 2021, where 98% of those who were aware of their HIV status were on treatment, and 96% of those on treatment were virally suppressed [51]. However, only 89% of adults aged 15 years and older knew their status, indicating there is still a gap in HIV testing [51]. To achieve the UNAIDS 95-95-95 targets by 2030 in Zambia, there is a need for interventions that enhance HIV testing.

## Study design and data source

This was a cross-sectional study that utilized secondary data from the couples recode file of the 2018 Zambia demographic and health survey (2018 ZDHS). The 2018 ZDHS was implemented by the Zambia Statistics Agency in partnership with the Ministry of Health, and data was collected from July 2018 to January 2019. This was the 6th Demographic and Health Survey conducted in Zambia since 1992 as part of several surveys obtained from the MEASURE DHS program, to provide reliable national estimates of demographic and health indicators such as gender relations, sexual and reproductive health, and other health issues relevant to the achievement of SDG's. The DHS is a reliable source of individual-level information on socio-economic characteristics, health, and development indicators in LMICs. Since 1999, the surveys have incorporated questions on women's empowerment that potentially allow for within and between countries comparisons using an intersectional lens [43]. The 2018 ZDHS included a nationally representative sample of 13,683 women aged between 15–49 years and 12,132 men aged between 15–59 years in 12,831 households, with a response rate of 96% among women and 92% among men. The sample design provided estimates at the national level, for both urban and rural areas, and each of the 10 provinces in the country.

## Sampling frame

The sampling frame used for the 2018 ZDHS is based on the Census of Population and Housing conducted in 2010. Zambia is divided into 10 provinces, and each province is subdivided into districts, each district into constituencies, and each constituency into wards. Additionally, wards are subdivided into convenient areas called census supervisory areas (CSAs) which are further subdivided into Standard Enumeration Areas (SEAs). The list of SEAs was used as the sampling frame for the 2018 ZDHS.

## Sampling

The survey used a stratified two-stage sample design. Each of the 10 provinces was stratified into urban and rural areas and this yielded 20 sampling strata. Samples of SEAs were selected independently from each stratum in two stages. In the first stage, 545 SEAs (198 Urban and 347 Rural) were selected with probability proportional to SEA size and with independent selection in each sampling stratum. The SEA size was the number of residential households residing in the SEA based on the sampling frame. A household listing was carried out in all the selected sample SEAs and the resulting lists of households served as the sampling frame for the second stage of sampling. In the second stage of sampling, a fixed number of 25 households per cluster was sampled from each cluster using equal probability systematic sampling from the newly created household listing. All women aged 15–49 years and men aged 15–59 years who were

usual members of the selected households or who spent the night before the survey in the selected households were eligible for the woman's questionnaire and the man's questionnaire, respectively. Our study was restricted to married/partnered Women with a focus on the couples recode section of the data since the SWPER index considers only married or partnered women. Women with incomplete information on any variable of interest were excluded from the analysis. A total of 13,683 women completed the survey, of which 5,560 were either married or living with a partner. Of the 5,560 coupled women, 5,328 had values to all variables of interest and thus were included in the analysis while 232 women had missing variables and were excluded. Therefore, this study analyzed data of 5,328 married/partnered women. Details of the sample selection are illustrated in Fig 1.

## Dependent variable

The dependent variable in this study was self-reported HIV testing, measured with two outcomes (yes/no). Women who responded "yes" to the question "Have you ever tested for HIV" were categorized as having tested for HIV and assigned a code of 1 while respondents who answered "no" to this question were categorized as never tested for HIV and assigned a code of 0 for this variable.

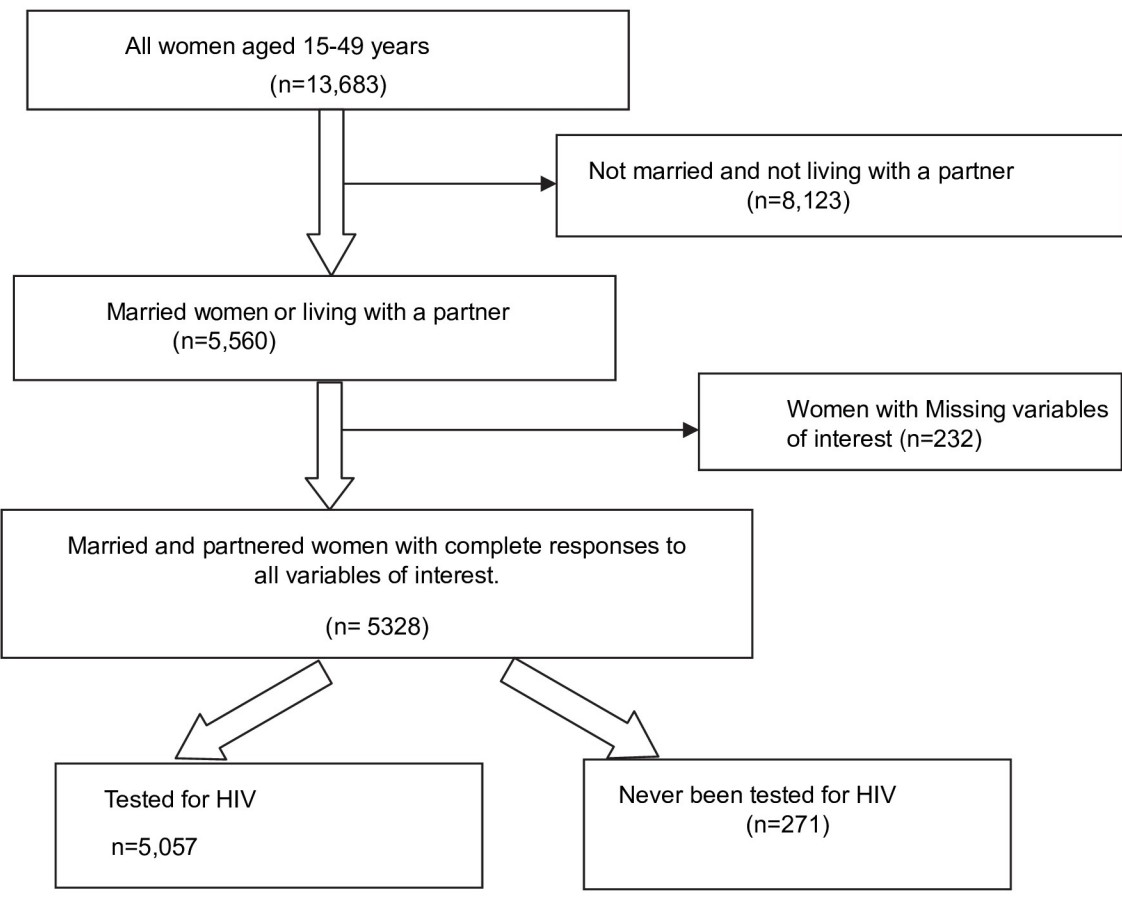

**Fig 1. Flow chart of sample selection.**

**Table 1. Items used in each domain of the SWPER index and corresponding response codes.**

| ITEM | CODE OR UNIT |
|---|---|
| **Attitude to violence domain** | |
| 1. Beating justified if wife goes out without telling husband | Yes = -1; Don't know = 0; No = 1 |
| 2. Beating justified if wife neglects the children | Yes = -1; Don't know = 0; No = 1 |
| 3. Beating justified if wife argues with husband | Yes = -1; Don't know = 0; No = 1 |
| 4. Beating justified if wife refuses to have sex with husband | Yes = -1; Don't know = 0; No = 1 |
| 5. Beating justified if wife burns the food | Yes = -1; Don't know = 0; No = 1 |
| **Social independence domain** | |
| 6. Frequency of reading newspaper or magazine | Not at all = 0, $<$once a week = 1, $\geq$once a week = 2 |
| 7. Woman education in completed years of schooling | |
| 8. Age of woman at first birth* | Years |
| 9. Age at first cohabitation | Years |
| 10. Age difference: woman's minus husband's age | Years |
| 11. Education difference: woman's minus husband's years of schooling | Years |
| **Decision Making Domain** | |
| 12. Who usually decides on respondents' health care | Husband or other alone = -1; Joint decision or respondent alone = 1 |
| 13. Who usually decides on large household Purchases | Husband or other alone = -1; Joint decision or respondent alone = 1 |
| 14. Who usually decides on visits to family or Relatives | Husband or other alone = -1; Joint decision or respondent alone = 1 |

\* This item age at first birth was imputed with age at first cohabitation for those women who had not had a child

**Source:** [43]

## Independent variables

Independent variables were categorized into socio-demographic and empowerment measures. The choice and categorization of these variables were based on data from existing studies [52–55]. Socio-demographic factors included in the study were participants' ages ($<$20, 20–29, 30–39, and 40–49 years), type of residence (Rural, Urban), education level collapsed into three categories (No education, primary, secondary or higher), Wealth index collapsed into three categories (poor, middle, rich), working status (working, not working), health insurance coverage (yes, no), and number of unions (once, more than once). Women's empowerment included three domains of the SWPER global index [43]. The SWPER global measures three domains of empowerment using 14 items from the DHS surveys. Five items are related to Women's opinion on justification of husbands beating their wives in specific situations (attitude to violence domain), six items relate to preconditions that enable women to achieve their goal (social independence domain) and three items relate to Women's participation in decision making (decision making domain) as indicated in (Table 1). All three measures of empowerment included all 14 items with different item weights for each empowerment measure. Standardization of the scores was done using the means and standard deviations for southern Africa. Each domain of the SWPER index was categorized into three groups as low empowerment, medium empowerment, and high empowerment based on set cut-offs as shown in (Table 2). Full details related to the equations used to calculate and standardize the SWPER scores are available online [30, 56]and described elsewhere [43].

**Table 2. Cut-offs used to categorize women's empowerment in each domain of the SWPER index.**

| CATEGORY | ATTITUDE TO VIOLENCE | SOCIAL INDEPENDENCE | DECISION-MAKING |
|---|---|---|---|
| Low Empowerment | $\leq$-0.700 | $\leq$-0.559 | $\leq$-1.000 |
| Medium Empowerment | >-0.700 $\leq$0.400 | >-0.559 $\leq$0.293 | >-1.000 $\leq$0.600 |
| High Empowerment | >0.400 | >0.293 | >0.600 |

**Source:** [56]

## Data analysis

Data was analyzed using SPSS version 22. Participants' socio-demographic characteristics, HIV testing rate, and levels of Women empowerment are presented using descriptive statistics. Bivariate analysis was conducted using the Chi-square test of independence to examine potential candidate variables for inclusion in logistic regression models. To account for the multistage sampling design used in the survey and produce estimates that were representative of the country, we employed the complex samples procedure in SPSS. Firstly, we set up a complex sampling plan using the CSPLAN command and adjusted for individual weights, clusters (primary sampling unit), and strata following the guidelines of DHS [57] on handling individual weight variables. Since our study involved a subpopulation of married/partnered women, we used the variable "current marital status" to specify the subpopulation during the analysis. We constructed eight logistic regression models. Model 1 was a univariable analysis of measures of women empowerment and socio-demographic factors with the dependent variable (ever tested for HIV). Variables that were statistically significant at $p<0.2$ in model 1 were analyzed simultaneously for their combined effects in model 2. Thereafter, models 3, 4, and 5 were constructed to assess the direction of the relationships between measures of empowerment (attitude to violence, social independence, and decision-making) respectively when considered independently, while adjusting for confounding socio-demographic variables that were statistically significant at $p<0.2$ from model 2. We then assessed the combined effects of measures of empowerment on HIV by running the variables simultaneously in model 6 while controlling for confounding socio-demographic factors. Finally, to understand if the influence of women's empowerment on HIV testing was independent of the measures in place to test for HIV in all women who attend antenatal care, we conducted separate logistic regression analyses among the two distinct groups of women: those who gave birth in the last five years and those who did not (models 7 and 8) respectively. The significance level was considered as $p <0.05$. We present both the AOR and the corresponding 95% CI.

## Ethics considerations

A formal request for analysis of all data was made to the DHS program, through their website (www.dhsprogram.com), and permission and access to the data was granted 30[th] November 2023. The original data was collected with ethical approval from the Tropical Disease and Research Center (TDRC) and the Research Ethics Review Board of the Center for Disease Control and Prevention (CDC) Atlanta. The study did not require any formal ethical approval because we used secondary data sources. The process of collecting data for the Zambia Demographic and Health Survey (ZDHS) necessitated obtaining consent from individuals aged 18 and above. Before seeking assent from minors, the DHS protocol mandated obtaining written informed consent from parents or guardians of all participants under 18 years of age.

## Results

Table 3 shows the background characteristics of the studied women. Overall, 5,328 married/partnered women were included in the analysis. The majority of women studied were between the ages of 20 to 39 years. Only 5.6% were aged below 20 years and 18.8% were aged between 40 to 49 years. Close to two-thirds (62.8%) were living in rural areas and more than half (52.2%) had primary education. With regards to wealth, a large proportion (41.6%) of women were poor, 20.2% were in the middle class and 38.2% were rich. More than two-thirds (85.2%) had no previous union. There was an almost equal proportion of women who were working and not working (49.5% and 50.5% respectively). The majority (97.8%) of women had no health insurance coverage. There was a high (94.9%) HIV testing rate among women.

**Table 3. Socio-demographic characteristics of married/partnered women in Zambia (n = 5328).**

| Variable | Frequency (Weighted) |
| --- | --- |
| | n (%) |
| **Ever been tested for HIV** | |
| Yes | 5057 (94.9) |
| No | 271 (5.1) |
| **Age group** | |
| <20 years | 300 (5.6) |
| 20 to 29 years old | 2158 (40.5) |
| 30 to 39 years old | 1871 (35.1) |
| 40 to 49 years old | 999 (18.8) |
| **Type of residence** | |
| Rural | 3346 (62.8) |
| Urban | 1982 (37.2) |
| **Level of education** | |
| No education | 510 (9.6) |
| Primary | 2781 (52.2) |
| Secondary or higher | 2037 (38.2) |
| **Wealth** | |
| Poor | 2218 (41.6) |
| Middle | 1078 (20.2) |
| Rich | 2032 (38.2) |
| **Number of unions** | |
| Once | 4542 (85.2) |
| More than once | 786 (14.8) |
| **Working status** | |
| Working | 2639 (49.5) |
| Not working | 2689 (50.5) |
| **Health insurance coverage** | |
| Covered | 119 (2.2) |
| Not covered | 5209 (97.8) |
| **Given birth in the last five years** | |
| Yes | 3912 (73.4) |
| No | 1416 (26.6) |

**Table 4. Bivariate association between socio-demographic characteristics and HIV testing among Zambian married/partnered women (N = 5328).**

| Socio-demographic characteristics | Tested for HIV | Never been tested for HIV. | Chi-Square (χ2) | P-value |
|---|---|---|---|---|
| | N = 5057 | N = 271 | | |
| | n (%) | n (%) | | |
| **Age group** | | | 113.3 | **<0.001** |
| <20 years | 251 (83.7) | 49 (16.3) | | |
| 20 to 29 years old | 2061 (95.5) | 96 (4.5) | | |
| 30 to 39 years old | 1820 (97.3) | 51 (2.7) | | |
| 40 to 49 years old | 925 (92.5) | 75 (7.5) | | |
| **Type of residence** | | | 56.1 | **<0.001** |
| Rural | 3118 (93.2) | 228 (6.8) | | |
| Urban | 1939 (97.8) | 43 (2.2) | | |
| **Level of education** | | | 138.4 | **<0.001** |
| No education | 435 (85.4) | 74 (14.6) | | |
| Primary | 2624 (94.4) | 157 (5.6) | | |
| Secondary or higher | 1998 (98.0) | 40 (2.0) | | |
| **Wealth** | | | 86.9 | **<0.001** |
| Poor | 2036 (91.8) | 182 (8.2) | 0.2 | |
| Middle | 1029 (95.4) | 49 (4.6) | | |
| Rich | 1992 (98.0) | 40 (2.0) | | |
| **Number of unions** | | | 0.1 | 0.781 |
| Once | 4312 (94.9) | 229 (5.1) | | |
| More than once | 745 (94.7) | 42 (5.3) | | |
| **Working status** | | | 6.7 | **0.034** |
| Working | 2573 (95.7) | 116 (4.3) | | |
| Not working | 2484 (94.1) | 155 (5.9) | | |
| **Health insurance coverage** | | | 5.2 | **0.006** |
| Covered | 118 (99.4) | 1 (0.6) | | |
| Not covered | 4939 (94.8) | 270 (5.2) | | |
| **Given birth in the last five years** | | | 52.7 | **<0.001** |
| Yes | 3765 (96.2) | 148 (3.8) | | |
| No | 1292 (91.3) | 123 (8.7) | | |

## Associations between socio-demographic characteristics and HIV testing

To assess associations between socio-demographic characteristics and the outcome variable (ever tested for HIV), the Pearson Chi-square test was conducted. Table 4 presents the results of the associations. There was no significant association between HIV testing and the number of unions a woman has had previously. However, significant associations were observed between HIV testing and age group, place of residence, level of education, wealth quintile, working status, health insurance coverage, and having given birth in the last five years preceding the survey.

## Women empowerment and HIV testing

As mentioned in previous sections, women's empowerment was measured using the standardized SWPER index. Three domains of empowerment were assessed, namely attitude to violence, social independence, and decision-making. To understand the influence of women empowerment on the utilization of HIV testing services, we conducted a Pearson Chi-square

**Table 5. Women empowerment and HIV testing among Zambian married/partnered women.**

| Women empowerment domain | Total n (%) | Tested for HIV | Chi-Square (χ2) | P-value |
|---|---|---|---|---|
| | N = 5328 | n (%) | | |
| | | N = 5057 | | |
| **Attitude to violence** | | | 17.2 | **0.002** |
| Low empowerment | 1669 (31.3) | 1553 (93.1) | | |
| Medium empowerment | 961 (18.1) | 920 (95.8) | | |
| High empowerment | 2698 (50.6) | 2584 (95.8) | | |
| **Social independence** | | | 62.5 | **<0.001** |
| Low empowerment | 1663 (31.2) | 1524 (91.6) | | |
| Medium empowerment | 2208 (41.5) | 2111 (95.6) | | |
| High empowerment | 1457 (27.3) | 1422 (97.6) | | |
| **Decision making** | | | 62.4 | **<0.001** |
| Low empowerment | 1226 (23.0) | 1118 (91.0) | | |
| Medium empowerment | 1286 (24.1) | 1212 (94.2) | | |
| High empowerment | 2816 (52.9) | 2729 (96.9) | | |

test between measures of empowerment and the outcome variable. The results of the analysis are displayed in (Table 5). More than half of women were highly empowered in the attitude to violence and decision-making domains of the SWPER index (50.6% and 52.9% respectively). However, a larger proportion (41.5%) of women had medium levels of empowerment in the social independence domain. There were significant associations between HIV testing and levels of empowerment in all three domains of the SWPER index.

## Women's empowerment and background characteristics

As shown in Table 6, women's empowerment indicators differ by background characteristics. Statistically significant correlates of women's empowerment in decision-making and social independence included age group, type of residence, level of education, wealth quintile, number of unions a woman has had, working status, health insurance coverage, and birth history in the past five years. Moreover, most socio-demographic characteristics had statistically significant associations with attitude to violence except the number of unions and working status.

## Predictors of HIV testing among Zambian coupled women

Complex samples binary logistic regression analysis was conducted to assess the relationship of the three measures of women empowerment and of socio-demographic characteristics with HIV testing. We constructed six models for the univariable and multivariable analysis. Model 1 focused on univariable analysis, examining the associations of socio-demographic factors and women empowerment measures with HIV testing individually, to identify candidate variables for inclusion in multivariable analysis (Table 7). Model 2 involved multivariable analysis, simultaneously adding all sociodemographic factors that were statistically significant at p<0.2 in model 1. Models 3 to 5 incorporated the three measures of empowerment individually, each introduced separately into the analysis while adjusting for socio-demographic factors that were statistically significant at p<0.2 in model 2. In Model 3, social independence and decision-making were excluded; in Model 4 attitude to violence and decision-making were excluded; and in Model 5 we excluded attitude to violence and social independence. Thus models 3, 4, and 5 addressed attitude to violence, social independence, and decision-making empowerment variables respectively while controlling for the effects of socio-demographic

**Table 6. Distribution of women's empowerment domains by background characteristics.**

| Socio-demographic characteristics | Decision making | | | | Social independence | | | | Attitude to violence | | | |
|---|---|---|---|---|---|---|---|---|---|---|---|---|
| | Low % | Medium % | High % | Chi-Square p-value | Low % | Medium % | High % | Chi-square p-value | Low % | Medium % | High % | Chi-square p-value % |
| **Age group** | | | | <0.001 | | | | <0.001 | | | | <0.001 |
| <20 years | 33.5 | 26.5 | 40.0 | | 53.8 | 42.9 | 3.3 | | 42.2 | 20.6 | 37.2 | |
| 20 to 29 years old | 25.9 | 21.1 | 53.0 | | 27.9 | 47.1 | 25.1 | | 33.3 | 18.9 | 47.8 | |
| 30 to 39 years old | 19.4 | 25.5 | 55.1 | | 31.9 | 36.2 | 31.8 | | 27.4 | 17.4 | 55.2 | |
| 40 to 49 years old | 20.3 | 27.5 | 52.2 | | 30.4 | 38.6 | 31.0 | | 31.2 | 16.6 | 52.2 | |
| **Type of residence** | | | | <0.001 | | | | <0.001 | | | | <0.001 |
| Rural | 29.5 | 24.5 | 46.0 | | 37.8 | 43.6 | 18.6 | | 37.4 | 17.5 | 45.1 | |
| Urban | 12.0 | 23.7 | 64.3 | | 20.0 | 37.9 | 42.1 | | 21.0 | 18.9 | 60.1 | |
| **Level of education** | | | | <0.001 | | | | <0.001 | | | | <0.001 |
| No education | 35.4 | 33.1 | 31.5 | | 70.7 | 20.8 | 8.5 | | 36.8 | 13.3 | 49.9 | |
| Primary | 27.0 | 25.2 | 47.8 | | 41.2 | 45.2 | 13.6 | | 37.8 | 18.3 | 43.9 | |
| Secondary or higher | 14.5 | 20.5 | 65.0 | | 7.7 | 41.5 | 50.8 | | 21.0 | 18.9 | 60.1 | |
| **Wealth** | | | | <0.001 | | | | <0.001 | | | | <0.001 |
| Poor | 31.8 | 25.7 | 42.5 | | 40.6 | 43.2 | 16.2 | | 43.1 | 16.8 | 40.1 | |
| Middle | 27.4 | 23.7 | 48.9 | | 35.4 | 46.2 | 18.4 | | 33.6 | 18.9 | 47.5 | |
| Rich | 11.1 | 22.7 | 66.2 | | 18.7 | 37.0 | 44.3 | | 17.3 | 18.9 | 63.8 | |
| **Number of unions** | | | | 0.030 | | | | <0.001 | | | | 0.405 |
| Once | 22.9 | 23.3 | 53.8 | | 29.1 | 42.3 | 28.6 | | 31.4 | 17.5 | 51.1 | |
| More than once | 23.9 | 29.0 | 47.1 | | 43.5 | 36.3 | 20.2 | | 31.0 | 21.3 | 47.7 | |
| **Working status** | | | | 0.001 | | | | 0.032 | | | | 0.942 |
| Working | 20.1 | 24.9 | 55.0 | | 30.0 | 40.8 | 29.2 | | 31.6 | 18.0 | 50.4 | |
| Not working | 26.0 | 23.4 | 50.7 | | 32.5 | 42.1 | 25.4 | | 31.0 | 18.1 | 50.9 | |
| **Health insurance coverage** | | | | <0.001 | | | | <0.001 | | | | <0.001 |
| Covered | 5.4 | 12.1 | 82.5 | | 0.9 | 15.5 | 83.6 | | 6.1 | 4.4 | 89.5 | |
| Not covered | 23.4 | 24.4 | 52.2 | | 31.9 | 42.0 | 26.1 | | 31.9 | 18.3 | 49.8 | |
| **Given birth in the last five years** | | | | 0.005 | | | | <0.001 | | | | 0.010 |
| Yes | 24.3 | 23.9 | 51.8 | | 31.0 | 43.1 | 25.9 | | 32.7 | 18.1 | 49.2 | |
| No | 19.1 | 24.8 | 56.1 | | 31.9 | 36.7 | 31.4 | | 27.4 | 17.9 | 54.7 | |

factors (Table 8). In model 6, all three empowerment measures were run simultaneously to determine their combined effects on HIV testing. The model explained 17.3% (Nagelkerke) of the variance in the dependent variable and correctly classified 94.9% of the cases.

When measures of women empowerment were assessed individually in the unadjusted model (model 1), high empowerment was significantly associated with HIV testing. Compared to women with low empowerment on attitude to violence, those with high empowerment and medium empowerment were 1.7 times (COR = 1.7; 95% CI: 1.1, 2.6) and 1.7 times (COR = 1.7; 95% CI: 1.7, 2.3) more likely to have been tested for HIV respectively. With regards to the social independence domain, women with high empowerment and those with medium empowerment were 3.8 times (COR = 3.8; 95% CI: 2.5, 5.8) and 2.0 times (COR = 2.0; 95% CI: 1.5, 2.7) more likely to have undergone an HIV test compared to those with low empowerment respectively. Moreover, women who were highly empowered and those with medium empowerment in decision-making were 3.1 times (COR = 3.1; 95% CI:

**Table 7. Univariable and multivariable analysis of socio-demographic factors associated with HIV testing among Zambian married/partnered women.**

| Variables | Model 1 (Univariable analysis) | | Model 2 | |
| --- | --- | --- | --- | --- |
| | COR (95% CI) | P value | AOR (95% CI) | P value |
| **Age group** | | | | |
| <20 years (ref) | 1.0 | | 1.0 | |
| 20 to 29 years | 2.7 (1.8–4.0) | **<0.001** | 2.4 (1.6–3.7) | **<0.001** |
| 30 to 39 years | 6.1 (3.9–9.5) | **<0.001** | 6.9 (4.6–10.4) | **<0.001** |
| 40 to 49 years | 3.6 (2.3–5.6) | **<0.001** | 5.1 (3.7–7.2) | **<0.001** |
| **Level of education** | | | | |
| No education (ref) | 1.0 | | 1.0 | |
| Primary | 2.8 (2.0–4.0) | **<0.001** | 2.8 (2.0–3.9) | **<0.001** |
| Secondary or higher | 8.6 (5.2–14.0) | **<0.001** | 5.9 (3.5–9.9) | **<0.001** |
| **Wealth** | | | | |
| Poor (ref) | 1.0 | | 1.0 | |
| Middle | 1.9 (1.4–2.6) | **<0.001** | 1.5 (1.1–2.2) | **0.006** |
| Rich | 4.5 (2.9–7.0) | **<0.001** | 2.4 (1.4–4.0) | **<0.001** |
| **Working status** | | | | |
| Not working (ref) | 1.0 | | 1.0 | |
| Working | 1.4 (1.0–1.9) | **0.034** | 1.2 (0.9–1.6) | 0.159 |
| **Number of unions** | | | | |
| Once (ref) | 1.0 | | - | - |
| More than once | 1.1 (0.7–1.5) | 0.781 | - | - |
| **Health insurance coverage** | | | | |
| Not covered (ref) | 1.0 | | 1.0 | |
| Covered | 9.7 (1.3–70.7) | **0.026** | 2.1 (0.3–16.8) | 0.491 |
| **Type of residence** | | | | |
| Rural (ref) | 1.0 | | 1.0 | |
| Urban | 3.3 (2.2–5.0) | **<0.001** | 0.8 (0.5–1.4) | 0.476 |
| **Given birth in the last five years** | | | | |
| No (ref) | 1.0 | | 1.0 | |
| Yes | 2.4 (1.8–3.3) | **<0.001** | 3.2 (2.4–4.4) | **<0.001** |
| **Attitude to violence** | | | | |
| Low empowerment (ref) | 1.0 | | | |
| Medium empowerment | 1.7 (1.1–2.6) | **0.022** | - | - |
| High empowerment | 1.7 (1.2–2.3) | **0.001** | - | - |
| **Social independence** | | | | |
| Low empowerment (ref) | 1.0 | | | |
| Medium empowerment | 2.0 (1.5–2.7) | **<0.001** | - | - |
| High empowerment | 3.8 (2.5–5.8) | **<0.001** | - | - |
| **Decision making** | | | | |
| Low empowerment (ref) | 1.0 | | | |
| Medium empowerment | 1.6 (1.2–2.2) | **0.004** | - | - |
| High empowerment | 3.1 (2.3–4.2) | **<0.001** | - | - |

COR-Crude Odds Ratio, AOR-Adjusted Odds Ratio

2.3, 4.2) and 1.6 times (COR = 1.6; 95% CI: 1.2, 2.2) more likely to have an HIV test done compared to those with low empowerment (model 1). After adjusting for the effects of other variables as illustrated in Table 8, women with high empowerment and medium empowerment in attitude to violence and social independence were no more likely to test for HIV than those

**Table 8. Associations between measures of women empowerment and HIV testing among Zambian married/partnered women.**

| Variables | Model 3 | | Model 4 | | Model 5 | | Model 6 | |
|---|---|---|---|---|---|---|---|---|
| | AOR (95% CI) | P value | AOR (95% CI) | P value | AOR (95% CI) | P value | AOR (95% CI) | P value |
| **Attitude to violence** | | | | | | | | |
| Low empowerment (ref) | 1.0 | | - | - | - | - | 1.0 | |
| Medium empowerment | 1.3 (0.8–2.0) | 0.245 | - | - | - | - | 1.4 (0.9–1.8) | 0.169 |
| High empowerment | 1.2 (0.9–1.7) | 0.165 | - | - | - | - | 1.2 (0.9–1.6) | 0.277 |
| **Social independence** | | | | | | | | |
| Low empowerment (ref) | - | - | 1.0 | | - | - | 1.0 | |
| Medium empowerment | - | - | 1.2 (0.9–1.6) | 0.218 | - | - | 1.3 (0.9–1.8) | 0.068 |
| High empowerment | - | - | 1.5 (0.9–2.4) | 0.146 | - | - | 1.6 (0.9–2.7) | 0.096 |
| **Decision making** | | | | | | | | |
| Low empowerment (ref) | - | - | - | - | 1.0 | | 1.0 | |
| Medium empowerment | - | - | - | - | 1.4 (0.9–1.9) | 0.053 | 1.4 (0.9–1.9) | 0.067 |
| High empowerment | - | - | - | - | 2.0 (1.5–2.8) | **<0.001** | 2.1 (1.5–2.9) | **<0.001** |
| **Age group** | | | | | | | | |
| <20 years (ref) | 1.0 | | 1.0 | | 1.0 | | 1.0 | |
| 20 to 29 years | 2.7 (1.8–4.0) | **<0.001** | 2.6 (1.7–3.9) | **<0.001** | 2.6 (1.7–3.9) | **<0.001** | 2.4 (1.6–3.6) | **<0.001** |
| 30 to 39 years | 6.1 (3.9–9.5) | **<0.001** | 5.8 (3.7–9.1) | **<0.001** | 5.8 (3.7–8.9) | **<0.001** | 5.3 (3.4–8.2) | **<0.001** |
| 40 to 49 years | 3.6 (2.3–5.6) | **<0.001** | 3.3 (2.1–5.3) | **<0.001** | 3.4 (2.2–5.2) | **<0.001** | 2.9 (1.9–4.7) | **<0.001** |
| **Level of education** | | | | | | | | |
| No education (ref) | 1.0 | | 1.0 | | 1.0 | | | |
| Primary | 2.8 (2.0–3.9) | **<0.001** | 2.6 (1.9–3.7) | **<0.001** | 2.6 (1.9–3.7) | **<0.001** | 2.4 (1.7–3.4) | **<0.001** |
| Secondary or higher | 5.9 (3.6–10.0) | **<0.001** | 4.9 (2.9–8.7) | **<0.001** | 5.3 (3.1–9.0) | **<0.001** | 4.1 (2.3–7.2) | **<0.001** |
| **Wealth** | | | | | | | | |
| Poor (ref) | 1.0 | | 1.0 | | 1.0 | | 1.0 | |
| Middle | 1.5 (1.1–2.2) | **0.009** | 1.6 (1.1–2.3) | **0.006** | 1.5 (1.1–2.2) | **0.012** | 1.5 (1.1–2.2) | **0.012** |
| Rich | 2.6 (1.7–4.1) | **<0.001** | 2.7 (1.7–4.3) | **<0.001** | 2.4 (1.5–3.9) | **<0.001** | 2.4 (1.5–3.7) | **<0.001** |
| **Working status** | | | | | | | | |
| Not working (ref) | 1.0 | | 1.0 | | 1.0 | | - | - |
| Working | 1.2 (0.9–1.7) | 0.140 | 1.2 (0.9–1.6) | 0.156 | 1.2 (0.9–1.6) | 0.200 | - | - |
| **Given birth in the last five years** | | | | | | | | |
| No (ref) | 1.0 | | 1.0 | | 1.0 | | 1.0 | |
| Yes | 3.3 (2.4–4.4) | **<0.001** | 3.2 (2.4–4.4) | **<0.001** | 3.4 (2.5–4.6) | **<0.001** | 3.3 (2.5–4.5) | **<0.001** |

AOR-Adjusted odds ratio

with low empowerment (models 3 and 4). The decision-making domain remained a significant predictor of HIV testing in the adjusted model (Models 5). Women who were highly empowered in decision-making were 2.0 times more likely to be tested for HIV than those with low empowerment in decision-making after adjusting for confounding socio-demographic variables (AOR = 2.0; 95% CI: 1.5, 2.8). However, there was no significant difference in HIV testing between those with medium and low empowerment in decision-making in the adjusted model. When considering the combined effects of all measures of empowerment and socio-demographic factors in model 6, women with high empowerment in decision-making were 2.1 times more likely to have tested for HIV than those with low empowerment (AOR = 2.1; 95% CI: 1.5, 2.9).

Among socio-demographic characteristics, age category, level of education, wealth, and having given birth in the last 5 years were the factors significantly associated with HIV testing.

The odds of HIV testing were higher among women aged 20 years or older than those aged below 20 years (AOR = 2.4; 95% CI: 1.6, 3.6 for 20–29 years old versus <20 years old, AOR = 5.3; 95% CI: 3.4, 8.2 for 30–39 years old versus <20 years old, AOR = 2.9; 95% CI: 1.9, 4.7 for 40–49 years old versus <20 years old). Compared to women with no education, the odds of HIV testing significantly increased with increasing level of education (AOR = 2.4; 95% CI: 1.7, 3.4 for primary versus no education, AOR = 4.1; 95% CI: 2.3, 7.2 for secondary or higher versus no education). With regards to wealth, the odds of HIV testing significantly increased with an increase in the level of wealth (AOR = 1.5; 95% CI: 1.1, 2.2 for middle wealth versus poor, AOR = 2.4; 95% CI: 1.5, 3.7 for rich versus poor). Moreover, women who gave birth in the last 5 years preceding the survey were significantly more likely to have tested for HIV than those who had not given birth (AOR = 3.3; 95% CI: 2.5, 4.5).

## Influence of recent childbirth on the relationship between women's empowerment and HIV testing behavior

Across all models (1–6), women who gave birth in the last five years preceding the survey were consistently found to have a significantly higher likelihood of HIV testing compared to those who did not. We, therefore, conducted separate logistic regression analyses among the two distinct groups of women: those who gave birth in the last five years and those who did not, to understand if the influence of women's empowerment on HIV testing is independent of recent childbirth. Model 7 assessed the influence of women's empowerment on HIV testing behavior among women who gave birth in the last five years and Model 8 assessed the influence of women's empowerment on HIV testing behavior among women who did not give birth in the last five years (Table 9). In both models, we controlled for confounding socio-demographic variables age group, level of education, and wealth. Results for the multivariate regression analyses among subgroups of women who gave birth in the last five years (model 7) and those who had no birth in the last five years (model 8) are similar to the regression analysis results of the full sample. Women who were highly empowered in decision-making were consistently more likely to be tested for HIV compared to those lowly empowered across both subgroups, similar to results observed in the full sample analysis. However, the odds of HIV testing among women highly empowered in decision-making were higher for the subgroup of women who had a birth in the last five years compared to those who had not (AOR 2.4 vs. 1.8) respectively.

Moreover, women in the middle and rich wealth categories had higher odds of HIV testing compared to those in the poor category among women who had not given birth in the last five years, similar to the trend observed in the full sample. However, among women who had given birth in the last five years, only the richest category was significantly more likely to have tested for HIV. The influence of education level and age group in determining HIV testing was similar in both subgroups of women.

## Discussion

A woman's ability to undergo HIV testing is significantly influenced by her level of empowerment. It is presumed that an empowered woman, whether through cultural, political, or professional means, possesses the confidence to independently choose to undergo HIV testing [3]. This autonomy ensures that she is not reliant on her husband or partner to make decisions regarding whether to undergo HIV testing. In Zambia, the cultural expectation of men's dominance and women's subordination has resulted in limited access to healthcare among women and reduced their autonomy to make independent decisions regarding their health [16, 17, 34]. Regardless, the influence of women empowerment on HIV testing has not been studied in Zambia. This study therefore examined the association between women empowerment and

**Table 9. Multivariate analysis on women's empowerment and HIV testing among subgroups of Zambian coupled women who gave birth and those who did not give birth in the last five years.**

| Variables | Model 7: Gave birth in the last five years. N = 3912 | | Model 8: No birth in the last five years N = 1416 | |
|---|---|---|---|---|
| | AOR (95% CI) | P value | AOR (95% CI) | P value |
| **Attitude to violence** | | | | |
| Low empowerment (ref) | 1.0 | - | 1.0 | - |
| Medium empowerment | 1.3 (0.7–2.2) | 0.441 | 1.5 (0.8–3.0) | 0.250 |
| High empowerment | 1.1 (0.7–1.7) | 0.612 | 1.3 (0.7–2.0) | 0.362 |
| **Social independence** | | | | |
| Low empowerment (ref) | 1.0 | - | 1.0 | - |
| Medium empowerment | 1.4 (0.9–2.0) | 0.150 | 1.2 (0.7–2.0) | 0.455 |
| High empowerment | 1.2 (0.6–2.5) | 0.545 | 1.9 (0.9–3.9) | 0.097 |
| **Decision making** | | | | |
| Low empowerment (ref) | 1.0 | - | 1.0 | - |
| Medium empowerment | 1.2 (0.8–1.9) | 0.333 | 1.5 (0.9–2.7) | 0.144 |
| High empowerment | 2.4 (1.5–3.6) | **<0.001** | 1.8 (1.1–3.0) | **0.015** |
| **Age group** | | | | |
| <20 years (ref) | 1.0 | - | 1.0 | - |
| 20 to 29 years | 2.3 (1.3–3.9) | **0.003** | 3.1 (1.5–6.5) | **<0.001** |
| 30 to 39 years | 4.8 (2.6–8.7) | **<0.001** | 7.4 (3.6–15.1) | **<0.001** |
| 40 to 49 years | 5.5 (2.8–10.8) | **<0.001** | 2.8 (1.5–5.2) | **0.001** |
| **Level of education** | | | | |
| No education (ref) | 1.0 | - | 1.0 | - |
| Primary | 2.9 (1.8–4.9) | **<0.001** | 2.6 (1.1–3.4) | **0.019** |
| Secondary or higher | 7.6 (3.6–16.3) | **<0.001** | 2.1 (0.9–5.1) | 0.087 |
| **Wealth** | | | | |
| Poor (ref) | 1.0 | - | 1.0 | - |
| Middle | 1.2 (0.8–1.8) | 0.423 | 2.1 (1.2–3.7) | **0.008** |
| Rich | 2.8 (1.3–5.9) | **0.009** | 2.5 (1.4–4.5) | **0.003** |

AOR-Adjusted odds ratio

HIV testing among married and partnered women in Zambia using the nationally representative Zambia demographic and health survey 2018 data. Three domains of women empowerment (attitude to violence, social independence, and decision-making) were measured using the novel, globally validated SWPER index. HIV testing among married and partnered women in Zambia was found to be encouragingly high, as 94.9% of women had undergone an HIV test. In Zambia, women are tested for HIV when pregnant during antenatal care attendance as part of HIV prevention, treatment, and care strategy [58]. This could explain the high HIV testing rate observed in this study. Delaying the detection of HIV and remaining uninformed about one's positive status entails various adverse consequences. Unawareness of HIV positivity raises the risk of transmitting the virus to others, including mother-to-child transmission for pregnant women [31, 59]. Additionally, late diagnosis can diminish the life expectancy of an infected person by elevating the viral load and diminishing the body's CD4+ T-cell count [60]. Timely identification of HIV is crucial to ensure that those with the infection receive appropriate treatment and care, thereby restricting the transmission of the virus to others [61].

The majority of women had higher levels of empowerment in decision-making and attitude to violence and lower levels of empowerment in social independence. This study found that women's empowerment influenced HIV testing. Women who were highly empowered in

attitude to violence, social independence, and decision-making were significantly more likely to have undergone an HIV test compared to those with medium or low empowerment in the unadjusted analysis. Moreover, decision-making was the strongest predictor of HIV testing. Women who were highly empowered in decision-making had higher odds of having tested for HIV compared to those with low empowerment, even after controlling for the effects of other variables in the adjusted model. These results are consistent with findings from a pooled analysis of 31 sub-Saharan African countries that assessed the association between HIV testing and the decision-making domain of the SWPER index [2]. However, attitude to violence and social independence domains had no significant associations with HIV testing after controlling for the effects of other variables in the adjusted models. Previous studies have reported mixed results on the relationship between partner violence and HIV testing, with some studies reporting negative associations [62–68], some reporting positive associations [69–72], and others finding no associations [73–78]. The observed differences in findings may stem from variations in study populations, contexts, and methodological approaches used in these studies. The cultural context of any society has been shown to be influential in shaping the social, economic, and political empowerment of women [79, 80]. Culture shapes societal norms, beliefs, and attitudes towards gender equality and women's empowerment initiatives [81]. Cultural factors such as traditions, customs, and values can either facilitate or hinder women's access to resources, decision-making power, and participation in public life [82].

Giving birth in the five years preceding the survey was a prominent confounding variable and consistent across all models. In all models 2–6, women who had given birth in the last five years were three times more likely to have tested for HIV compared with women who had not given birth. In Zambia, HIV testing is done routinely during pregnancy as part of services offered during antenatal care attendance. We, therefore, conducted separate logistic regression analyses among the two distinct groups of women: those who gave birth in the last five years and those who did not, to further understand if the influence of women's empowerment on HIV testing was independent of the measures in place to test for HIV in all women who attend antenatal care. We anticipated that empowerment might not play a significant role in influencing HIV testing behavior among women who recently gave birth, as testing in this scenario is often routine and may demand less motivation. Conversely, we anticipated that testing outside of antenatal care might require more initiative and determination. While attitude to violence and social independence did not exhibit any significant influence on HIV testing behavior in the two subgroups of women after controlling for other variables, decision-making emerged as the most influential empowerment variable across all models, regardless of recent childbirth. These results echo findings from a study conducted in Tanzania [3]. This underscores the significance of women's ability to make independent decisions regarding their health, including the decision to undergo HIV testing. Regardless of measures in place to test all women attending antenatal care in Zambia, women who possess greater decision-making power are more likely to take proactive steps toward HIV testing, emphasizing the importance of promoting empowerment initiatives that enhance women's decision-making power.

Finally, among socio-demographic factors studied, age category, level of education, wealth quintile, and having given birth in the last 5 years were significantly associated with HIV testing across all models. Regarding age category, women who were aged 20 years or more had higher odds of testing for HIV compared to those younger than 20 years. Similar results have been reported in prior studies [2, 83–85]. A possible explanation for this consistent finding could be attributed to the fact that this age group is less likely to have gone through pregnancy compared to older age categories, hence they may have had fewer opportunities to encounter HIV testing through routine antenatal care, resulting in lower overall odds of testing for HIV within this age group. In many African settings, routine HIV testing is commonly integrated

into antenatal care services for pregnant women. In Zambia, persons under the age of 16 require parental consent to access HIV testing services, unless they are married, pregnant, or parents themselves [86, 87]. This requirement may act as a barrier for younger individuals to seek HIV testing independently, particularly if they do not meet the criteria for exceptions such as being married, pregnant, or a parent. As a result, older individuals, who are not subject to the same parental consent requirement, may have greater autonomy to access HIV testing services, leading to the observed higher odds of testing among this age group compared to younger individuals. Women with primary education and those with secondary education or higher were consistently more likely to test for HIV compared to those with no education across all models. This finding was consistent with studies conducted in Ethiopia [88, 89], Nigeria [90], and Tanzania [91]. Women with some form of education are more likely to be aware of the importance of HIV testing and have a greater understanding of the risks associated with HIV and the benefits of early detection [92]. Moreover, educational attainment is linked to socioeconomic status, and individuals with higher socioeconomic status may face fewer barriers, such as financial constraints or lack of awareness, that could impede access to HIV testing [93]. Enabling Zambian women to attend at least primary education would aid in improving HIV testing uptake.

Additionally, wealth was an independent predictor of HIV testing. Rich women and those in the middle wealth quintile had higher odds of testing for HIV compared to poor women. This is supported by different studies in Tanzania [3], and in the Gambia [85]. Giving birth in the last 5 years preceding the survey remained significantly associated with HIV testing, even after controlling for the effects of other variables. Similar results were reported in another similar study among Tanzanian women [3]. These results could be attributed to the fact that antenatal care services integrate various health interventions, including HIV testing, to provide comprehensive care to pregnant women. As a result, women who gave birth in the last 5 years were more likely to have undergone HIV testing as part of routine antenatal care, leading to higher odds of HIV testing in this demographic group.

## Implications for policy and practice

The findings of this study underscore the need for targeted interventions to improve HIV testing rates among partnered women in Zambia. Women's decision-making power is a crucial factor influencing their engagement in HIV testing. Addressing this requires a multifaceted approach that not only empowers women but also challenges existing cultural norms and barriers. This involves designing programs that provide education and resources to help women understand their health rights and the importance of HIV testing. Additionally, community engagement initiatives that address cultural norms and attitudes toward women's empowerment could facilitate broader societal change. By challenging the cultural expectations of male dominance and promoting equitable decision-making within households, we can create an environment that encourages women to take proactive steps regarding their health. Furthermore, targeted outreach programs that specifically aim to educate younger women about HIV testing and provide accessible resources could address the barriers they face, particularly concerning parental consent laws. Creating supportive spaces for discussion and education about HIV can empower women to make informed choices about HIV testing, regardless of their age or relationship dynamics. Finally, collaboration with local health services to ensure that HIV testing is routinely integrated into all health care services, not just antenatal care, could enhance accessibility for all women, thereby improving overall testing rates.

### Strengths and limitations of the study

The large sample size in this study enhanced the statistical robustness of this investigation. Furthermore, the utilization of weight application and complex samples plan during analysis addressed potential biases and considerations associated with the DHS sampling design, resulting in unbiased national estimates. However, this study recognizes some limitations. Firstly, we could not infer causality due to the cross-sectional design of the study. Secondly, the analysis involved only married and partnered women because questions of the SWPER index are addressed to this category of women. Hence, results cannot be generalized to the broader category of Zambian women. Moreover, many empowered women may not be married or may plan to marry later in life. Additionally, disabled women and sex workers, who are among the most marginalized and disempowered groups, may be less inclined to be married, thus rendering them ineligible for inclusion in the study. Furthermore, our findings cannot be generalized to adolescents, many of whom are unmarried. We recognize the potential implications of this restriction and the need for further research to explore the empowerment of women who are not married or do not have a partner. Participants were asked about HIV testing that could have happened at any point in their lifetime(ever tested for HIV) with the potential disconnect between historical testing behavior and the present empowerment context. The findings from this study should, therefore, be interpreted within the context of these limitations.

## Conclusion

This study sheds light on the critical influence of women's empowerment on HIV testing among married and partnered women in Zambia. While overall HIV testing rates were high, empowered women, particularly those highly empowered in decision-making, were more likely to have undergone HIV testing. The study emphasizes the influence of contextual factors, such as routine antenatal care testing during pregnancy, on HIV testing patterns. Moreover, socio-demographic factors, including age, education, wealth, and recent childbirth, were identified as significant predictors of HIV testing.

## Supporting information

**S1 Checklist. STROBE checklist for *cross-sectional studies* adapted to our study.**
(DOCX)

## Acknowledgments

The authors express gratitude to the DHS program for providing the data used in the analysis.

## Author Contributions

**Conceptualization:** Whiteson Mbele.

**Formal analysis:** Whiteson Mbele.

**Methodology:** Whiteson Mbele, Andreas Ndapewa Frans.

**Software:** Whiteson Mbele.

**Supervision:** Phyllis Dako-Gyeke.

**Visualization:** Andreas Ndapewa Frans.

**Writing – original draft:** Whiteson Mbele.

**Writing – review & editing:** Phyllis Dako-Gyeke, Jean Claude Ndayishimiye, Jordanne Ching.

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
