## [Decision Letter · Decision Letter 0]

13 May 2024

PGPH-D-23-02543

Women empowerment and utilization of HIV testing among couples in Zambia: Evidence from the Zambia Demographic and Health Survey 2018

Dear Dr. Mbele,

Thank you for submitting your manuscript to PLOS Global Public Health. After careful consideration, we feel that it has merit but does not fully meet PLOS Global Public Health’s publication criteria as it currently stands. Therefore, we invite you to submit a revised version of the manuscript that addresses the points raised during the review process.

Please note that we have only been able to secure a single reviewer to assess your manuscript. We are issuing a decision on your manuscript at this point to prevent further delays in the evaluation of your manuscript. Please be aware that the editor who handles your revised manuscript might find it necessary to invite additional reviewers to assess this work once the revised manuscript is submitted. However, we will aim to proceed on the basis of this single review if possible.

The reviewer suggested that you provide more information on the theory of empowerment beyond the index used, and explain the limitations of this index with regard to their results and the context of the study.

We look forward to receiving your revised manuscript.

Kind regards,

Emma Campbell, Ph.D

Staff Editor

Journal Requirements:

1. We ask that a manuscript source file is provided at Revision. Please upload your manuscript file as a .doc, .docx, .rtf or .tex.

Reviewers' comments:

Reviewer's Responses to Questions

**Comments to the Author**

1. Does this manuscript meet PLOS Global Public Health’s publication criteria? Is the manuscript technically sound, and do the data support the conclusions? The manuscript must describe methodologically and ethically rigorous research with conclusions that are appropriately drawn based on the data presented.

Reviewer #1: Yes

2. Has the statistical analysis been performed appropriately and rigorously?

Reviewer #1: Yes

3. Have the authors made all data underlying the findings in their manuscript fully available (please refer to the Data Availability Statement at the start of the manuscript PDF file)?

Reviewer #1: Yes

4. Is the manuscript presented in an intelligible fashion and written in standard English?

Reviewer #1: Yes

5. Review Comments to the Author

Reviewer #1: Title: Women empowerment and utilization of HIV testing among couples in Zambia: Evidence from the Zambia Demographic and Health Survey 2018.

General commentary :

This study is original, relevant and well presented. My main comments suggest that the authors provide more information on the theory of empowerment beyond the index used, and explain the limitations of this index with regard to their results and the context of the study.

Introduction :

Line 76 : “Empowerment has been variably defined, with limited data for

77 comparisons across countries, especially in Sub-Saharan Africa”: based on all the evidence you described, what is your definition of womens’empowerment in this study ?

Methodology:

Line 123: “the country recorded a drop of 43.7% in HIV testing in 2020 compared to 2019”: are there differences between men and women?

Line 168: “Our study was restricted to married/partnered Women with a focus on the couples recode section of the data since the SWPER index considers only married or partnered women”.: That's a very important point. You could also discuss the restriction of the empowerment index to women who are married or in a couple: is the empowerment of women essentially linked to the fact of being married or having a partner? What about women who are not married or do not have a partner?

Table 2 and reference 42 : the link to your reference does not work to see the details of the equations used to calculate SWPER

Tables 1 and 2 : mention the sources above or below the tables

Results:

Table 3: Since pregnant women are tested for HIV, perhaps you could carry out a sensitivity test, taking into account women who have not given birth in the last five years or who have never given birth? It could be interesting to see if the influence of women's empowerment on access to HIV testing is independent of the measures put in place for those who are pregnant. For example, will a woman in a partnered relationship go for a screening test on her own if she is not pregnant or does not have children or has not had children in the last 5 years ? My comment is also echoing to the following sentences in the discussion (line 410-415): “The diminished significance of associations with HIV testing after adjusting for the effects of other variables may be attributed to a potential confounding factor. It is plausible that the higher proportion of women in our study who had given birth in the last 5 years could have influenced the results. In Zambia, HIV testing is done routinely during pregnancy as part of services offered during antenatal care attendance, and this might have played a role in shaping women’s decision to test for HIV”.

Line 275: “(Table 4) presents the results of the associations”: delete the parenthesis

Discussion :

Line 419: “The observed differences in findings may stem from variations in study populations, contexts, and methodological approaches used in these studies.” Previous studies on empowerment theory have indicated that empowerment is a reality dependent on context and culture. You could therefore go further and discuss this point in your results.

Line 428: Aren't there other possible reasons: the age at which it is possible to have a screening test without parental consent? Sexuality education for young people? easier access to screening for young people without children?

Perhaps you could include an appendix describing the characteristics of women according to their level of empowerment. This would give a more detailed idea of the women concerned by each situation and could also help you to comment more effectively on the effect of other characteristics by making the link with empowerment.

You might consider adding a grid adapted to your study, based on the recommendations for reporting quantitative studies: in the absence of a guideline for analysing secondary data, it seems to me that the STROBE guide could be useful to you: https://www.equator-network.org/reporting-guidelines/coreq/.

6. PLOS authors have the option to publish the peer review history of their article (what does this mean?). If published, this will include your full peer review and any attached files.

**Do you want your identity to be public for this peer review?** For information about this choice, including consent withdrawal, please see our Privacy Policy.

Reviewer #1: No

---

## [Decision Letter · Decision Letter 1]

2 Oct 2024

PGPH-D-23-02543R1

Women empowerment and utilization of HIV testing among couples in Zambia: Evidence from the Zambia Demographic and Health Survey 2018

Dear Dr. Mbele,

Thank you for submitting your manuscript to PLOS Global Public Health. After careful consideration, we feel that it has merit but does not fully meet PLOS Global Public Health’s publication criteria as it currently stands. Therefore, we invite you to submit a revised version of the manuscript that addresses the points raised during the review process.

I regret that it was again not possible to secure an Academic Editor to handle your manuscript. To ensure a thorough evaluation, we have consulted with an additional reviewer, and their comments are provided in the attachment. They have a number of thoughtful suggestions to improve the contextualization of your study and the validity of your conclusions and recommendations. They also request additional details about the analysis.

We look forward to receiving your revised manuscript.

Kind regards,

Marianne Clemence

Staff Editor

Journal Requirements:

Additional Editor Comments (if provided):

Reviewers' comments:

Reviewer's Responses to Questions

**Comments to the Author**

1. If the authors have adequately addressed your comments raised in a previous round of review and you feel that this manuscript is now acceptable for publication, you may indicate that here to bypass the “Comments to the Author” section, enter your conflict of interest statement in the “Confidential to Editor” section, and submit your "Accept" recommendation.

Reviewer #1: All comments have been addressed

Reviewer #2: (No Response)

2. Does this manuscript meet PLOS Global Public Health’s publication criteria? Is the manuscript technically sound, and do the data support the conclusions? The manuscript must describe methodologically and ethically rigorous research with conclusions that are appropriately drawn based on the data presented.

Reviewer #1: Yes

Reviewer #2: Yes

3. Has the statistical analysis been performed appropriately and rigorously?

Reviewer #1: Yes

Reviewer #2: Yes

4. Have the authors made all data underlying the findings in their manuscript fully available (please refer to the Data Availability Statement at the start of the manuscript PDF file)?

Reviewer #1: Yes

Reviewer #2: Yes

5. Is the manuscript presented in an intelligible fashion and written in standard English?

Reviewer #1: Yes

Reviewer #2: Yes

6. Review Comments to the Author

Reviewer #1: I would like to thank the authors for their responses to my comments. Just 2 comments: Line 229: “We constructed six logistic regression models”: the authors should correct, they now have 8 models, if I am not mistaken.

Line 270: It seems a bit strange to start a sentence with brackets. It may be done, but if it's possible to start the presentation of tables directly without putting them in brackets, except in the middle or at the end of a sentence, it would be more reader-friendly.

Thanks for the description of the empowerment score. It's interesting to see that the higher women's level of education and the more they are rich, the higher their level of empowerment. These characteristics also play a role in HIV testing uptake. This seems to show, at least in part, that to act on women's level of empowerment on the one hand, and on their HIV testing uptake on the other, it is fundamental to fight for women's education and against socio-economic inequalities.

Reviewer #2: My reviewer commets are attached to the review.

7. PLOS authors have the option to publish the peer review history of their article (what does this mean?). If published, this will include your full peer review and any attached files.

**Do you want your identity to be public for this peer review?** For information about this choice, including consent withdrawal, please see our Privacy Policy.

Reviewer #1: No

Reviewer #2: No

---

## [Decision Letter · Decision Letter 2]

22 Nov 2024

Empowerment and utilization of HIV testing among partnered women in Zambia: Evidence from the Zambia Demographic and Health Survey 2018

PGPH-D-23-02543R2

Dear Mbele,

We are pleased to inform you that your manuscript 'Empowerment and utilization of HIV testing among partnered women in Zambia: Evidence from the Zambia Demographic and Health Survey 2018' has been provisionally accepted for publication in PLOS Global Public Health.

Best regards,

Ferdinand C Mukumbang, PhD

Academic Editor

Reviewer Comments (if any, and for reference):

Reviewer's Responses to Questions

**Comments to the Author**

1. If the authors have adequately addressed your comments raised in a previous round of review and you feel that this manuscript is now acceptable for publication, you may indicate that here to bypass the “Comments to the Author” section, enter your conflict of interest statement in the “Confidential to Editor” section, and submit your "Accept" recommendation.

Reviewer #1: All comments have been addressed

Reviewer #3: All comments have been addressed

2. Does this manuscript meet PLOS Global Public Health’s publication criteria? Is the manuscript technically sound, and do the data support the conclusions? The manuscript must describe methodologically and ethically rigorous research with conclusions that are appropriately drawn based on the data presented.

Reviewer #1: Yes

Reviewer #3: Yes

3. Has the statistical analysis been performed appropriately and rigorously?

Reviewer #1: Yes

Reviewer #3: Yes

4. Have the authors made all data underlying the findings in their manuscript fully available (please refer to the Data Availability Statement at the start of the manuscript PDF file)?

Reviewer #1: Yes

Reviewer #3: Yes

5. Is the manuscript presented in an intelligible fashion and written in standard English?

Reviewer #1: Yes

Reviewer #3: Yes

6. Review Comments to the Author

Reviewer #1: I would like to thank the authors for answering my questions. I have no further questions.

Reviewer #3: The manuscript is well written and thoroughly referenced. My only issue is the currency of the data and while you had the data you could have explored association with initiation of anti-retroviral medication and viral suppression which may be affected by empowerment.

7. PLOS authors have the option to publish the peer review history of their article (what does this mean?). If published, this will include your full peer review and any attached files.

**Do you want your identity to be public for this peer review?** For information about this choice, including consent withdrawal, please see our Privacy Policy.

Reviewer #1: No

Reviewer #3: **Yes: **Kesetebirhan Delele Yirdaw
